# High Mortality and Graft Loss after Infective Endocarditis in Kidney Transplant Recipients: A Case-Controlled Study from Two Centers

**DOI:** 10.3390/pathogens10081023

**Published:** 2021-08-13

**Authors:** Yanis Tamzali, Clément Danthu, Alexandra Aubry, Romain Brousse, Jean-François Faucher, Zhour El Ouafi, Pierre Rufat, Marie Essig, Benoit Barrou, Fatouma Toure, Jérôme Tourret

**Affiliations:** 1Kidney Transplantation Department, Assistance Publique—Hôpitaux de Paris APHP, Pitié-Salpêtrière Hospital, FR-75013 Paris, France; 2Kidney Transplantation Departement, Limoges University Hospital, Inserm Umr 1092, Resinfit, FR-87000 Limoges, France; Clement.DANTHU@chu-limoges.fr (C.D.); zhour.elouafi@yahoo.fr (Z.E.O.); 3Department of Bacteriology and Hygiene, Sorbonne Université, Assistance Publique—Hôpitaux de Paris APHP, Pitié-Salpêtrière Hospital, (Cimi-Paris), Inserm U1135, FR-75013 Paris, France; alexandra.aubry@aphp.fr; 4Department of Nephrology and Dialysis, Sorbonne Université, Assistance Publique—Hôpitaux de Paris APHP, Tenon Hospital, FR-75019 Paris, France; romain.brousse31@gmail.com; 5Infectious Diseases and Tropical Medicine Department, Limoges University Hospital, INSERM, University Limoges, IRD, U1094, Institute of Epidemiology and Tropical Neurology, GEIST, FR-87000 Limoges, France; jean-francois.faucher@unilim.fr; 6Département D’information Médicale (DIM), Sorbonne Université, Assistance Publique—Hôpitaux de Paris APHP, Pitié-Salpêtrière Hospital, FR-75013 Paris, France; pierre.rufat@aphp.fr; 7Nephrology Department, Université Paris Saclay, Assistance Publique—Hôpitaux de Paris APHP, Ambroise Paré Hospital, FR-92100 Boulogne Billancourt France, CESP Inserm 1018, FR-94800 Villejuif, France; marie.essig@aphp.fr; 8Kidney Transplantation Department, Sorbonne Université, Assistance Publique—Hôpitaux de Paris APHP, Pitié-Salpêtrière Hospital, INSERM UMR 1082, FR-75013 Paris, France; benoit.barrou@aphp.fr; 9Department of Nephrology, Transplantation and Dialysis, University Hospital of Limoges, INSERM, CNRS UMR7276, U1262, CRIBL, FR-87000 Limoges, France; fatouma.toure@chu-limoges.fr; 10Kidney Transplantation Department, Sorbonne Université, Assistance Publique—Hôpitaux de Paris APHP, Pitié-Salpêtrière Hospital, INSERM UMR 1138, FR-75013 Paris, France; jerome.tourret@aphp.fr

**Keywords:** infective endocarditis, kidney transplantation, survival analysis, graft failure, transplant infectious diseases

## Abstract

Kidney transplant recipients (KTRs) tend to develop infections with characteristic epidemiology, presentation, and outcome. While infective endocarditis (IE) is among such complications in KTRs, the literature is scarce. We describe the presentation, epidemiology, and factors associated with IE in KTRs. We performed a retrospective case/control study which included patients from two centers. First episodes of definite or possible IE (Duke criteria) in adult KTRs from January 2010 to December 2018 were included, as well as two controls per case, and followed until 31 December 2019. Clinical, biological, and microbiological data and the outcome were collected. Survival was studied using the Kaplan–Meier method. Finally, we searched for factors associated with the onset of IE in KTRs by the comparison of cases and controls. Seventeen cases and 34 controls were included. IE was diagnosed after a mean delay of 78 months after KT, mostly on native valves of the left heart only. Pathogens of digestive origin were most frequently involved (six *Enterococcus spp*, three *Streptococcus gallolyticus*, and one *Escherichia coli*), followed by *Staphylococci* (three cases of *S. aureus* and *S. epidermidis* each). Among the risk factors evaluated, age, vascular nephropathy, and elevated calcineurin inhibitor through levels were significantly associated with the occurrence of IE in our study. Patient and death-censored graft survival were greatly diminished five years after IE, compared to controls being 50.3% vs. 80.6% (*p* < 0.003) and 29.7% vs. 87.5% (*p* < 0.002), respectively. IE in KTRs is a disease that carries significant risks both for the survival of the patient and the transplant.

## 1. Introduction

Infective endocarditis (IE) is an invasive infection characterized by high inoculum of a pathogen that has a strong propensity to form biofilms, and that is also capable of systemic dissemination. Worldwide, this severe disease (with 30% 1-year mortality) remains rare. Over the last decades, a trend was observed with an increase in staphylococcal infections and increasing incidence in older patients with more comorbidities [1].

End-stage renal disease (ESRD) is a growing worldwide concern, with almost one million ESRD patients in the United States in 2019 [2], mostly elderly patients with comorbidities. IE in ESRD patients has already been described in chronic hemodialysis (HD) patients, where it may be a consequence of a staphylococcus bacteriemia, an extremely common condition in this patient subgroup exposed to dialysis fistula puncture or central catheter usage three times a week. Solid organ transplant recipients (SOTRs) undergo immunosuppressive treatment to prevent graft rejection, and as a result they are susceptible to more frequent, more severe, and atypical infections [3]. IE in this population is not well described in the literature apart from uncommon and isolated cases.

We therefore sought to describe the presentation, epidemiology, risk factors, and outcome of IE in kidney transplant recipients (KTRs) through a case/control study.

## 2. Material and Methods

### 2.1. Study Design, Setting, and Participants

We performed a retrospective case–control study with patients from two centers: the Pitié-Salpêtrière (Assistance Publique—Hôpitaux de Paris, France) and in the Dupuytren (Limoges, France) Hospitals. Cases were screened from the databases of the medical-based information systems in the two hospitals. Patients were included according to the following conditions: kidney transplant recipients with a functioning allograft, diagnosed for a first episode of certain or possible IE, on a native heart (exclusion of kidney-heart recipients), between January 2010 and December 2019. Two controls were included with each case; these were the patients who had received a kidney transplant just before and just after the case in the same center, provided that those patients survived at least until the delay of IE in the case. If one of the following exclusion criteria applied to the controls, the next-previous or the next-after transplanted patient was included instead: diagnosis of IE during the study period (included as a case), death before the diagnosis of the IE in the corresponding case, or presence of another organ transplanted with the kidney.

### 2.2. Clinical Data and Definitions

The diagnosis of definite or possible IE was made according to the modified DUKE criteria [1,4].

The onset date of the episode was defined by the start of antibiotic therapy for IE. The following clinical data were collected from the medical record:
Medical history: presence of a heart disease at high risk of IE heart (prosthetic valves, congenital cyanotic heart disease, or history of infective endocarditis), a history intravenous drug use, pre-transplant diabetes or new onset diabetes after transplantation (NODAT).Kidney transplantation (KT) history: the most recent estimated glomerular filtration rate (eGFR, MDRD formula) considered as stable before IE onset, induction and maintenance immunosuppressive treatments before and at the time of the infectious episode, the presence of high levels of calcineurin inhibitors or antimetabolites prior to the infectious episode (trough level > 10 ng/mL for tacrolimus or >150 ng/mL for cyclosporine, mycophenolate mofetil area under the curve (MMF AUC) > 60 mg.h/L), the treated episodes of rejection and viral infections (BK virus and cytomegalovirus, CMV) between transplantation and the IE episode.The characteristics of the IE with the time to onset after KT, bacteriological documentation, infectious gateway, ultrasonography features, type of valve, vascular (embolization, intracranial hemorrhages, mycotic aneurysms) and immunological (glomerulonephritis) complications, and the presence of an indication for surgery according to the European Society of Cardiology [5],IE therapeutic management: antibiotic therapy used, treatment duration and surgical managementOutcome: patient and renal graft survival one year after the IE were collected. For controls, the delay between the IE diagnosis in the corresponding case and the event (death, loss of graft function, loss to follow-up, or end of the study) was considered for the survival analysis. The end of the study was 31 December 2019.

### 2.3. Statistical Analysis

The annual incidence was estimated by dividing the annual number of cases of IE by the number of living KTRs in the two centers during the same year.

Quantitative variables are presented as mean ± standard deviation (all data were normally distributed). Comparisons were made using a Student’s t-test. Qualitative variables are presented as numbers (percentages). The data were compared using the Fischer or Chi2 test.

Risk factors for IE were searched using univariate logistic regression, including all clinical characteristics differently distributed between cases and controls with a *p* < 0.1. A two-sided *p* value < 0.05 was considered statistically significant in the univariate analysis. We did not perform multivariate analysis, as the validity criterium of at least 10 events per included variable was not met.

Survival analyses were performed using the Kaplan–Meier method. A log-rank test was performed for the comparison between the two groups, with a *p* value < 0.05.

The statistical analyses were performed using GraphPad PRISM^®^ (GraphPad Software, San Diego, CA, USA) and Stata^®^ (StataCorp LLC, College Station, TX, USA).

## 3. Ethics

All patients provided consent to be included in the local databases before their transplantation and granted us the authorization to anonymously use their clinical data in the perspective of clinical research. The clinical databases were approved by the French Ethics Committee on the Treatment of Computerized Data in the Field of Medical Research, under the auspices of the French Ministry of Research (declaration number: 2097646.v.0 for Pitié-Salpêtrière Hospital and 2210609609.v.0 for Limoges Hospital).

## 4. Results

### 4.1. Population and Incidence of IE

Over the study period 17 KTRs were diagnosed with IE, resulting in a mean annual incidence of 1.1%. We identified and included 34 controls. The characteristics of the population are shown in Table 1. IE occurred mostly in men (sex ratio 2:1), after a mean delay of 77.8 ± 82.3 months after KT. However, men predominated among KT controls in a non-significantly different proportion. The mean age at IE diagnosis was 63.8 ± 13 years. All cases were recipients of a first kidney transplant.

### 4.2. Clinical Presentation and Microbiological Epidemiology of IE

The clinical presentation and management of the cases of IE are summarized in Table 2. All cases were left heart endocarditis, and only three occurred on prosthetic valves. Mitral and aortic valves were equally involved.

The bacteria causing the infection were: *Enterococcus spp* (6), *Streptococcus gallolyticus* (3), coagulase-negative *Staphylococci* (CNS; 3), *Staphylococcus aureus* (3), *Escherichia coli* (1), and one undocumented case. Concerning IE on prosthetic valves (one mechanical and two bioprosthetic valves), *Streptococcus gallolyticus*, *Enterococcus faecalis,* and *Staphylococcus epidermidis* were found (one of each). One patient with a bioprosthetic valve had a definite *Staphylococcus epidermidis* IE, followed 10 months later by a new episode with *Escherichia coli* (not included in the analysis). The microorganisms were considered to be of digestive origin in 10 cases and of cutaneous origin in 6 cases. There was no case of IE with an oral origin.

Vascular embolism was observed in six subjects (Appendix A). No immunological complications were found.

Seven patients presented with an indication for surgery as recommended by the European Society of Cardiology in 2015 [5] (two abscesses, two severe regurgitations, and three vegetations > 15 mm), three underwent surgery (one with severe aortic and mitral regurgitations and two with a vegetation > 20 mm).

The duration of the antibiotic therapy was six weeks, except for one case in which the patient received only four weeks of gentamicin and daptomycin for a methicillin-resistant *S. aureus* uncomplicated endocarditis of a native valve. Aminoglycosides were used in twelve patients (four with Staphylococcal IE, three with Enterococcal IE, three with Streptococcal IE, and one with *E. coli* IE and in the undocumented case). Vancomycin was used in three patients (one case of *Enterococcus faecium*, methicillin-resistant *Staphylcococcus epidermidis,* and *Staphylococcus aureus* each).

## 5. Analysis of Risk Factors

In order to identify potential risk factors for IE, the distribution of clinical characteristics, comorbidities, and IS protocols (Table 1) were compared between cases and controls. Only age and the initial nephropathy were significantly different. Cases were significantly older than controls (63.8 ± 13 vs. 55.6 ± 12 years, *p* = 0.03, Table 1), and suffered more frequently from a vascular nephropathy than (29.4% vs. 5.9%, *p* < 0.01 for the global comparison of all nephropathies). 

IS protocols were evenly distributed between cases and controls; only high trough levels of CNI tended to be observed more frequently in cases than in controls (*p* = 0.08; Table 1).

We then performed a logistic regression including these potential risk factors (age, high CNI trough levels, and initial nephropathy as a bimodal variable: vascular nephropathy vs. other). In the univariate analysis age (OR: 1.06 per year 95% CI (1.01–1.12), *p* = 0.04), elevated CNI through levels (OR: 7.22 95% CI (1.19 to 44.0), *p* = 0.03), and vascular nephropathy (OR: 8.72 95% CI (1.53–49.8)) appeared as risk factors for IE.

## 6. Patient and Graft Survival

Patient survival was greatly diminished after an episode of IE (Figure 1A). One- and five-year survival were 58 (95% CI: 31–77) vs. 100 and 50 (95% CI: 24–72) vs. 80.6 (95% CI: 59–92) for cases and controls, respectively (*p* < 0.003). The median survival was 20 months for cases and was not reached in controls at the end of the study. Eight patients with IE died, including seven within the first year of follow-up, and five within the first six months.

Death-censored graft survival was also greatly lower in cases than in controls (Figure 1B). The estimated graft survival at one and five years was 81.5 [44–95] months vs. 100 and 29.7 [1.6–70] vs. 87.5 [65–96] for cases and controls, respectively (*p* < 0.002). The median graft survival for cases was 49 months and was not reached at the end of follow-up for controls.

The characteristics of the cases complicated by death or allograft loss (without death over the study period) are summarized in the Appendix A. 

## 7. Discussion

IE is well described in the general population [1,6,7], with an annual incidence in Western countries of 40 cases per million people [8,9]. There are numerous studies in hemodialysis (HD) patients, where IE is reported as both frequent (annual incidence of 1.7 to 2%) [10,11] and deadly, with a 45.6% mortality [12,13]. This is probably due to the frequent vascular punctures in these patients, which represents the most common infectious gateway [12,13].

In our study, the annual incidence is 1.1‰ and surprisingly high if we consider that none of the patients in our study harbored a central venous device nor underwent repeated vascular puncture as a risk factor. This is probably the consequence of the IS treatment and of the comorbidities of this population.

IE in KTRs occurred mostly in younger people than in the general population [1,6,7,11], and on hearts without any IE-predisposing conditions, as in the general population [14]. In our series, IE only occurred on left heart valves, but right heart valve IE has also been reported in KTRs previously [14].

The microbiology also appears different from what has been described in HD patients, where there is a large predominance of *Staphylococci* [12,13]. In that respect, the epidemiology in our study is closer to that of the general population [1]. We found that digestive bacteria predominate, with *Enterococci* as the most frequent pathogen, as previously described in SOTRs [14,15,16,17,18,19,20,21]. A hypothetical mechanism, in the absence of an identified digestive gateway (colonoscopy was performed in most of the cases where a digestive bacterium was identified, even though we did not collect the results of this exam in our study), could be the alteration of the gut microbiota by the combination of antibiotic treatments frequently used after transplantation, and of immunosuppressive drugs. The changes in the gut flora after transplantation often include an increase in proteobacteria, and an increase in the Firmicutes/Bacteroidetes ratio [22,23,24]. This dysbiosis has been associated with the development of infections due to immune dysregulation and/or the promotion of virulent strains [3,25].

In contrast with what has been reported for SOTRs in one study [21,26], but consistent with what was found in another [21], we found no case of fungal endocarditis. The undocumented cases in our study survived for more than seven years after an empiric antibiotic treatment, making a fungal origin of the infection highly unlikely.

The time to onset after transplantation was consistent with other reports for KTRs, with a mean delay of 3 to 5 years [21,27].

Interestingly, our study identified the combination of immunosuppression and vascular disease as risk factors for IE in KTRs. Age (already reported as a risk factor for IE in KTRs in [1]) and chronic kidney disease are associated with both conditions. Univariate analysis also identified high CNI trough levels (probably an additional argument in favor of the role of immunosuppression) and vascular nephropathy (an indirect indicator of the role of global vascular disease).

A history of CMV infection was not found to be associated with IE, as opposed to earlier studies where co-occurring CMV replication was found to be associated with both disease and mortality [20,28], suggesting that CMV infection is a hallmark of profound immunosuppression.

We found IE is a frequently deadly disease, with a one-year mortality of 43%, when all the controls survived after the same delay. This mortality appears comparable with what has previously been described in SOTRs [17,20] and in HD patients [17,20], but is much higher than what has been described in the general population [7]. This enhanced severity could reflect the frailty of immuno-compromised hosts but could also be related to the virulence of the pathogens or favored by the gut microbiota dysbiosis, as mentioned above. Inadequate treatment, due to imprecise GFR estimation tools in KTRs [29] and under- or over-dosing in the anti-infectious therapy, could also contribute to this poor outcome. Importantly, only three of the seven patients with a consensual surgical indication actually received surgery. Among the eight patients who died, six had a consensual surgical indication and only one actually received surgery. We did not collect the reasons for surgery refusal. It is possible that the global burden of comorbidities (IS, chronic kidney disease, other comorbidities associated with KT) was considered a contraindication.

More than 70% of the patients received aminoglycosides. According to the 2015 ESC Guidelines [5], the remaining indications for aminoglycoside are Enterococcal IE, some Streptococcal IE (two weeks of a gentamycin-based regimen in young patients with normal renal function), and IE without microbiological documentation [5]. Only 41% of the aminoglycoside prescriptions in our study match one of these indications. The overuse of aminoglycoside can probably be explained by the fact that most of the cases were included before 2015.

Finally, death-censored graft survival after IE also appeared unexpectedly low, with half of the grafts lost after four years, versus 13% in the controls after the same delay. Endocarditis is known to result in acute and long-term kidney complications [30] due to infectious, post-infectious, immunological, or drug-related mechanisms. Our study was not designed to explore the cause of renal failure in the patients who were included. Antibiotic-related renal toxicity (70% of the patients received either vancomycin or aminoglycosides therapy) or immunosuppression modulation following the episode may have participated to this poor prognosis. The potential consequences of nephrotoxic drugs on long-term kidney function should particularly be considered when taking decisions in antibiotic treatments in this specific setting.

IE after kidney transplantation is a critical disease, both for the patient and the transplant function. Bacteria from the gut microbiota are overly represented, which underlines the fact that the host–pathogen crosstalk is insufficiently understood and should be further investigated in KTRs.

## Figures and Tables

**Figure 1 pathogens-10-01023-f001:**
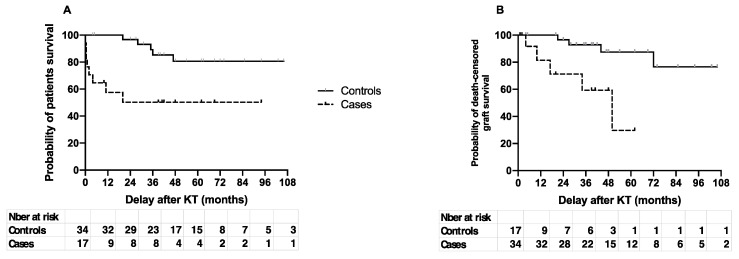
Patient (**A**) and death-censored graft (**B**) survival. Cases (dotted line) and controls (solid line) survival were estimated according to the Kaplan–Meier method.

**Table 1 pathogens-10-01023-t001:** Characteristics of the patients and comparison of risk factors for IE between cases and controls.

	Cases, N = 17 * n (%) or Mean ± SD	Controls, N = 34 * n (%) or Mean ± SD	*p* Value
Demographics			
Age (years)	63.8 ± 13.4	55.6 ± 11.7	0.03
Sex (male)	11 (64.7)	19 (55.9)	0.54
Comorbidities			
Intravenous drug use	0 (0)	0 (0)	1
Heart prosthetic valve	3 (17.6)	2 (5.9)	0.32
Diabetes	7 (41)	7 (20.6)	0.2
- Pre-existing	4 (23.5)	4 (11.8)	0.21
- After transplant	3 (17.6)	3 (8.8)	0.35
Initial nephropathy			<0.01
Vascular/Hypertension	5 (29.4)	2 (5.9)	
Diabetes	3 (17.6)	5 (14.7)	
PKD	3 (17.6)	4 (11.8)	
IgA nephropathy	2 (11.8)	2 (5.9)	
Glomerulonephritis	2 (11.8)	3 (8.8)	
aHUS	2 (11.8)	0	
Undetermined	0	7 (20.6)	
Other	0	11 (32.4)	
Transplant features			
First transplantation	17 (100)	30 (88.2)	0.29
CMV infection	9 (52.9)	11 (32.3)	0.26
BK virus infection	1 (5.9)	8 (23.5)	0.24
Treatment of acute rejection	2 (11.8)	3 (8.8)	1
Induction therapy			
ATG	13 (76.5)	24 (70.6)	1
Basiliximab	4 (23.5)	10 (29.4)	1
Maintenance therapy			
Steroids	15 (88.2)	29 (85.3)	1
MMF	16 (94.1)	30 (88.2)	0.65
CNI	17 (100)	34 (100)	1
Other	1 (5.9)	3 (8.8)	1
Drug monitoring			
AUC MMF > 60 mg·h/L	1/6 (16)	0/10 (0)	0.36
Elevated CNI trough level **	5/14 (35.7)	2/22 (9.1)	0.08
IE features			
Time for onset after KT (months)	77.8 ± 82.3	NA	1
Last available eGFR before IE (mL/min/1.73 m^2^)	43.6 ± 21.9	52.3 ± 24.0	0.14

aHUS: atypical hemolytic uremic syndrome; ATG: anti-thymocyte globulin; AUC: area under the curve; CMV: cytomegalovirus; CNI: calcineurin inhibitors; eGFR: estimated glomerular filtration rate; KT: kidney transplantation. MMF: mycophenolate mofetil; NA: not applicable; PKD: polycystic kidney disease. *: Except for drug monitoring data where the number of available data is specified in the table **: >10 ng/mL for tacrolimus and 150 ng/mL for ciclosporin.

**Table 2 pathogens-10-01023-t002:** Clinical, radiological, and microbiological characteristics of IE in KTRs.

Characteristics	N = 17, n (%) Unless Otherwise Specified
Definite IE	12 (70.6)
Possible IE	5 (29.4)
Valve	
Native	14 (82.4)
Prosthetic	3 (17.6)
Aortic IE	5 (39.4)
Mitral IE	7 (41.2)
Mitral and aortic	4 (23.5)
Echocardiography data	
No vegetation	2 (11.8)
Ring abscess and/or severe valve leakage	6 (35.3)
Vascular complications	6 (36.3)
Microbiology	
*Enterococci*	6 (35.3)
*Streptococcus gallolyticus*	3 (17.6)
*Staphylococcus aureus*	3 (17.6)
Coagulase-negative Staphylococci	3 (17.6)
*Escherichia coli*	1 (5.9)
No documentation	1 (5.9)
Probable origin of the causative bacterium	
Digestive	10 (58.8)
Cutaneous	7 (41.2)
Unknown	1 (5.9)
Treatment	
Antibiotic treatment duration (weeks), mean ± SD	5.9 ± 0.5
Aminoglycoside use	12 (70.6)
Indication for surgery	7 (41.2)
Surgery	3 (17.6)

IE: infective endocarditis.

## Data Availability

The data and material are available upon request to the corresponding author.

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
