# Peer review of "High Mortality and Graft Loss after Infective Endocarditis in Kidney Transplant Recipients: A Case-Controlled Study from Two Centers"

_pathogens, 2021, doi:10.3390/pathogens10081023_

Round 1
Reviewer 1 Report
The study is interesting since the data about IE in KTR are scarce. The manuscript is well-written.
I have the following comments:
- In the abstract, the authors wrote that the starting date of the study was 2007, but in the main text, they wrote that the starting date was 2010. Please correct.
- Regarding the DUKE criteria, if the terms Definite and Possible IE are explained in the revised manuscript, it would be nice to note these criteria. Otherwise, there is no need to explain the terms Definite and Possible IE in the text.
- In the statistical analysis, the authors noted that a Student t-test was used to compare means. What test was used for the skewed variables?
- In the results, the effect of age and vascular nephropathy was repeated in two separate points. Please omit the one.
- The authors noted that they evaluated eGFR one year after the IE. However, in the results, only data about the patient and death-censored graft survival are provided.
- Was the initial eGFR included in the analysis of the predisposing factors and the outcome? Besides IS treatment and gut dysbiosis, CKD is a well-known immunosuppressive condition.
Reviewer 2 Report
The authors performed a retrospective, case-control study on 51 KTR to evaluate features, epidemiology, risk factors and outcomes in patients with infective endocarditis (IE) after KT. In 17 patients with IE, the disease was diagnosed at a mean time of 78 months after KT, IE occurred mostly on native valves and on heart valves only. Enterococcus spp. was found in 35% of cases. Patients with IE were significantly older and had a tendency for elevated calcineurin inhibitors level. This is a topic of interest and previous data are scarce. Below are my comments and recommendations:
- Case-control matching. Was patients’ matching performed also for age, gender or other variables? If so, this should me mentioned at the method section.
- Risk factors. One of the objectives of the present study was to identify risk factors for IE. Analyzed variables were presented in table 1 and table 3. The authors performed only a comparative analysis and found that patients with IE were significantly older and tended to have elevated calcineurin inhibitors level. I suggest a superior analysis should be performed to identify risk factors (e.g. Cox regression analysis) if this is possible. If not, I think age and CNI levels should be described as characteristics of the patients and not as risk factors. In this last scenario, the objective of risk factors should be reconsidered and included in that of features of the patients.
- Table 1 and 3 information. Table 1 contains comparative details regarding patients’ characteristics and table 3 about comorbidities, immunosuppressive treatment and drug monitoring. Data from those two tables should be attached to form only one table that should be structured for more clarity and to avoid duplicate information as: demographics (age, gender), comorbidities (diabetes, intravenous drug use, heart prosthetic valve), causes for CKD/ initial nephropathy, transplant features (first transplantation, CMV infection, BKV infection, treatment of AR, immunosuppressive treatment, drug monitoring), IE features (time of onset, eGFR at diagnosis).
- Diabetes in Table 1 and 3. The data presented in these tables, regarding diabetes are discordant.
Table 1- 7 pts with IE: 4 with pre-existing DM (23.5%) and 3 with NODAT (17.5%)
7 control pts: 3 with pre-existing DM (8.8%) and 4 with NODAT (11.8%)
Table 3- 7 pts with IE: 4 pre-existing DM (23.5%) and 3 with NODAT (23.1%)
7 control pts: 4 pre-existing DM (11.8%) and 3 with NDOAT (10%)
- Are there any data regarding complement level, urinalysis or indication for kidney biopsy at the moment of IE?
- Abbreviations. If the authors decided to explain the abbreviations under the table, this should be done for all abbreviations.
- Complications. Authors found that 7 patients developed vascular embolism. Data regarding germ type, involved valve and management should be provided. Was there any case of sepsis? These should also be mentioned.
- Patient and graft survival
- the description of patients who have lost their graft or died needs to be improved with details related to germ type, involved valve, sepsis, type of treatment and surgery indication.
- figure 1: the y axis title should be changed to “probability of patient survival”, and the numbers from the section “Nber at risk”, in the survival table under the graph, should be deleted because they cause confusion, being in fact the months of post-transplant follow-up.
-figure 2: the y axis title should be changed to “probability of death-censored graft survival”, and the numbers from the section “Nber at risk” should also be deleted.
- Conclusion. Some of the conclusions cannot be supported by the results. The authors claimed that “The incidence of IE after kidney transplantation is higher than in the general population, but lower than in hemodialysis patients’’. The study was not designed to compare the incidence of IE from KTR with patients from HD or from the general population. Therefore, this conclusion has no support. The authors should focus on findings related to the epidemiology obtained, features associated with KTR, with IE, microbiological particularities and patient and graft survival outcomes.
Round 2
Reviewer 1 Report
The authors addressed all issues.
Reviewer 2 Report
The authors responded elegantly and efficiently to all recommendations. The new version of the manuscript is improved and I think it has enough quality now. I have no other suggestions.